# Mosquito Egg Raft Distribution Is Affected by Semiochemicals: Indication of Interspecific Competition

**DOI:** 10.3390/insects15050364

**Published:** 2024-05-16

**Authors:** Nimrod Shteindel, Yoram Gerchman, Alon Silberbush

**Affiliations:** 1Faculty of Natural Sciences, University of Haifa, Haifa 3103301, Israel; nimrod.sh@gmail.com (N.S.); gerchman@research.haifa.ac.il (Y.G.); 2Oranim College of Education, Tivon 3600600, Israel

**Keywords:** mosquitoes, egg rafts, oviposition habitat selection, competitor-released signals, semiochemicals, crowding signals

## Abstract

**Simple Summary:**

Interspecific competition occurs when two or more species require similar resources. Competition could be avoided by selecting a habitat with fewer competitors. While this behavior is well known, the identification mechanism is poorly understood. Mosquitoes select larval habitats during oviposition, and high competitor densities reduce larval survival. In this study, we show that ovipositing mosquito females can detect and avoid pools containing crowding signals originating from interspecific larvae. Furthermore, when larvae were not crowded, the habitat was found to be attractive to conspecifics. These findings increase our understanding of signals affecting mosquito oviposition, competitor recognition, and habitat selection under competition conditions.

**Abstract:**

Numerous species of animals alter their behavior in response to increasing competition. To do so, they must possess the ability to detect the presence and density of interspecific competitors. We studied the role of semiochemicals released by increasing densities of larval *Culiseta longiareolata* Macquart on female oviposition habitat selection in two field experiments. Similarly to *C. longiareolata* larvae, subordinate *Culex laticinctus* Edwards are periphyton grazers who dwell in rain-filled pools in the Mediterranean region. We show that *C. laticinctus* females oviposited significantly less in mesocosm pools that were treated with crowding signals originating from *C. longiareolata* larvae. In the second experiment, we placed a similar number of larvae directly inside the 50 L mesocosms. These low-density mesocosms did not affect *C. laticinctus* oviposition but were attractive to conspecific oviposition. These results increase our understanding of the female ability to detect species-specific signals, indicating increased larval competition.

## 1. Introduction

Interspecies competition is a reciprocally negative interaction between populations of two or more species [1,2]. This fundamental interaction between species that share a similar niche is considered to be one of the most important factors in the shaping of the ecological community [2,3], as well as being an important driver of evolutionary speciation [4]. The high cost of competition is associated with several alterations in animal behavior. In the presence of a competing species, individuals alter their foraging activity [2,5,6] in addition to producing aggressive interference and shifts in mating success [6,7]. Habitat selection is another mechanism affected by competition, and some species shift their activity, in space (to a less desirable habitat) or in time (to a different time of day or a different season) to reduce competitive interactions [8]. These behavior alterations happen in response to increased densities of both populations and thus require the ability to identify competitor presence as well as an estimation of conspecifics [9]. Although several studies have demonstrated the ability of competitors to do just that [8,10,11], the mechanism of identification is usually overlooked.

Chemical signals, or semiochemicals, are an important source of information for numerous animal species. This is especially true in aquatic systems, since odorants tend to travel better in water compared to auditory and visual signals [12]. Aquatic species alter their behavior in response to semiochemicals indicating a food source, predator presence, or conspecific density [13]. With few exceptions, e.g., [14], studies on behavior alteration occurring as a result of the semiochemicals associated with interspecific competitors looked at mere recognition and overlooked the density-dependent effects.

Ovipositing mosquito females (Diptera: Culicidae) are excellent models for the study of semiochemical effects on habitat selection. Mosquitoes are characterized by a complex life cycle where adults are free to range the landscape, but the immature are confined to the aquatic habitat where they hatched. Female mosquitoes provide little parental care beyond the selection of an appropriate oviposition site, making oviposition a critical factor in larval survival. Ovipositing females are attracted to several bacteria-released semiochemicals that are associated with nutrients for future larvae [15,16]. Females also use semiochemicals to detect predators and avoid oviposition in sites where predation risk is high [16,17]. In addition, gravid females can quantify predators [18] and conspecific larvae [19], rather than simply being aware of their presence. In this study, we examined the ability of ovipositing mosquito females to detect and respond to the presence and density of semiochemicals originated by the larvae of a dominant competitor species.

### Study Species

*Culiseta longiareolata* Macquart are highly abundant throughout the Mediterranean region [20]. The females typically oviposit in small, often temporary, rain-filled bodies of water and are often the earliest colonizers of these habitats following rain [21,22,23]. Because these rain-filled pools are both ephemeral and limited in number, they are a valuable resource to amphibians and aquatic insects. The larvae of another mosquito species, *Culex laticinctus* Edwards, are often associated with *C. longiareolata* breeding sites [20,24]. The larvae of the latter are considerably smaller in comparison to *C. longiareolata* larvae (Figure 1). *Culiseta longiareolata* larvae are considered herbivorous and feed mainly on periphyton algae and bacteria [25]. Nevertheless, fourth instar *C. longiareolata* larvae are considered to be highly aggressive competitors of other freshwater species, such as *Bufo virdis* tadpoles [26] and other mosquito larvae [27]. This aggressive behavior towards other aquatic dwellers may result in the death of larvae of other mosquito species [28,29], as well as vertebrates such as *Bufo virdis* tadpoles [26].

The purpose of this study was to examine the effects of the chemical signals produced by high densities of the dominant competitor *C. longiareolata* larvae on ovipositing mosquito females. The study was conducted with field mesocosms that mimic the rain-filled pools that are the natural larval habitats of both species. We hypothesized that water with high *C. longiareolata* larval density would contain crowding signals. These signals, indicating high competition, will be associated with larval habitats of poor quality. We therefore predict that ovipositing females will avoid these habitats. In a second experiment, we examined the effects of the actual larvae who were not subjected to crowding. According to the ideal free distribution theory, conspecifics usually prefer habitats with low densities. The exceptional cases are habitats with densities that are close to zero. In these cases, Allee’s principal states that the suitability of a habitat may actually increase with increasing density to a certain point [30]. Subordinate competitor species are expected to avoid competition and reduce oviposition in habitats containing interspecific competitors.

## 2. Methods

### 2.1. Field Experiments

Field experiments were conducted at the Oranim college campus botanical gardens Tivon-Israel, 32°42′47″ N 35°06′30″ E, between May and June 2023, a period that shows a peak in activity for ovipositing mosquitoes in that area. We monitored mosquito oviposition in black plastic pools sized 66.04 × 50.8 × 15.24 cm^3^. The pools, mimicking natural breeding sites, were organized in a randomized block design, containing eight blocks of three treatments (*n* = 24) randomly distributed within each block. Pools within a block were placed ~1 m apart and blocks were spaced ~10 m from each other. Pools were filled with ~50 L tap water and supplemented with 10 g of rodent chow to enhance oviposition. We collected mosquito egg rafts daily from the water surface of each pool. The collected egg rafts were hatched, and the larvae were raised to the 4th instar and their species were identified using [20].

Experiment 1—Conditioned water with crowded larvae: We produced conditioned water with highly crowded *C. longiareolata* larvae by placing 200 4th instar larvae in 400 mL plastic cups for 24 h. This density of 500 larvae/L is considered high but it is not an untypical density for this species [22,26]. A second treatment with medium-crowded larvae included 20 larvae in 400 mL (50 larvae/L); a third set included control cups with no larvae. Larvae were fed with ~0.05 g of finely grounded fish flakes (Sera vipan, 42.2% crude protein) that were added to each cup. Larvae were removed from the water daily using a fine net, and the dead and pupated were replaced. The conditioned water (without the larvae) was then added to the experimental pools each day at sunset (the beginning of mosquito activity). All pools were emptied and refilled every 5 days in order to reduce the effect of accumulating factors such as bacteria, algae, debris, etc.

Experiment 2—Live larvae experiment: We used the same design as in the previous experiment. Here, we placed 2 densities of living *C. longiareolata* larvae directly in the field mesocosms. The high-density treatment included 200 4th instar larvae (~number of larvae originated from a single egg raft [21,31]). A medium density treatment consisted of 20 larvae per pool, and the third pool was a control pool without larvae. Dead and pupated larvae were replaced daily, and all pools were emptied and refilled every 5 days.

### 2.2. Statistical Analysis

We used the total number of all egg rafts per pool, collected for each mosquito species across all dates, as a dependent variable. We used square root transformations of these values with an addition of 0.5 to all values, to homogenize among-treatment variance [32]. The homogeneity of variance was then tested using Levene’s test. We conducted separate univariate ANOVAs for each mosquito species in each of the experiments using “Block” and “Treatment” as fixed factors. Treatment means were compared using Tukey–Kramer honest significant difference test (HSD) when the main effect of treatment had *p* < 0.1, using α = 0.05 for individual HSD comparisons. All analyses used SPSS statistics for Windows version 24 [33].

## 3. Results

The first field experiment ran for 15 days (3–18 May 2023) and the second for 25 days (22 May–16 June 2023). During these periods, we collected a total of 335 and 340 egg rafts from both setups, respectively. One of the blocks in the first field experiment contained an especially low number of egg rafts in the control pool. This block was removed from the analysis. All the egg rafts collected in both of the field experiments belonged to one of three mosquito species: *Culiseta longiareolata* (Macquart), *Culex laticinctus* (Edwards), and *Culex pipiens* (Linnaeus). The egg rafts of these three species appeared in similar amounts and consisted of 34.3%, 34%, 31.6%, 25.6%, 40.9%, and 33.5% for *C. longiareolata*, *C. laticinctus*, and *C. pipiens* during the first and second experiments, respectively.

In the first field experiment, the three species showed significantly different responses to the conditioned water treatments. *Culiseta longiareolata* egg raft distribution was not affected by the treatments (F_2,12_ = 0.1; *p* = 0.9; Figure 2a). By contrast, *C. laticinctus* oviposition was significantly affected by the different treatments (F_2,12_ = 5.25; *p* = 0.02), with significantly fewer egg rafts oviposited in pools containing conditioned water of either highly crowded or medium-crowded larvae (Figure 2b). The oviposition distribution pattern of *C. pipiens* was not affected by the treatments (F_2,12_ = 0.39; *p* = 0.69; Figure 2c).

In the second field experiment with live larvae in the pools, *C. longiareolata* oviposition showed a dramatic response to the presence of larvae (F_2,14_ = 10.64; *p* = 0.002), with many more egg rafts found in the high-density pools in comparison to the medium-density or control pools without larvae (Figure 3a). The egg raft distribution of both *C. laticinctus* and *C. pipiens* was not significantly affected by the presence of *C. longiareolata* larvae (F_2,14_ = 0.14; *p* = 0.87 and F_2,14_ = 0.89; *p* = 0.43, respectively, Figure 3b,c).

## 4. Discussion

This study focused on the role of semiochemicals indicating increasing competition in habitat selection. We hypothesized that water containing highly crowded *C. longiareolata* larvae (20 and 200 larvae in 400 mL) would contain chemical signals. These semiochemicals are likely to be specific to *C. longiareolata* larvae, and they are likely to accumulate in pools as a result of larval crowding. In this case, the pool will be associated with highly dense *C. longiareolata* larvae and avoided by ovipositing female even after it is diluted with fresh water [34]. The origin of these semiochemicals may not necessarily be the larvae themselves. For example, a gravid mosquito female detects and avoids predatory backswimmers via volatile hydrocarbons released from the predator’s cuticle [35]. Larvivorous fish, on the other hand, are detected via signals associated with symbiotic bacteria [36]. Regardless of the semiochemicals’ origin, the hypothesis that ovipositing mosquitoes respond to specific chemical cues is strongly supported by our results. *Culiseta longiareolata* females did not respond to conspecific crowding signals (Figure 2a). However, a similar number of conspecific larvae that were not crowded were found to be attractive to gravid *C. longiareolata* females (Figure 3a). Conspecific density does not necessarily cause an immediate decrease in habitat suitability. A low number of conspecifics may be favorable to colonizers over an empty habitat [30]. This trend was shown for ovipositing mosquitoes in response to increasing conspecific larvae [19] and eggs [37]. It is suggested that a habitat with low conspecific density may indicate site persistence, potential mates, and overall appropriate conditions without increased competition. Higher larval densities will cause reduction in habitat suitability and should therefore be avoided during oviposition. The observed lack of conspecific response may point to the high conspecific density tolerated by this species [22,26].

By contrast to conspecifics, the presence of interspecific-dominant competitors reduces habitat quality even at low densities [8,38,39]. Mosquito larvae are often confined to the oviposition site until metamorphosis. The presence of interspecific larvae of a dominant competing species at this site will often result in reduced survival [40]. Even if survival to metamorphosis is not significantly reduced, interspecific competition results in other factors associated with a decline in population size, such as reduced adult body size, longevity, or changes in time to metamorphosis [41]. In our case, competition can be completely avoided by placing the larvae in a competition-free habitat during oviposition. Our results show that females of the subordinate *C. laticinctus* preferred to oviposit in pools that lacked water that were conditioned with crowded larvae of the dominant *C. longiareolata* larvae (Figure 2b). Similar numbers of larvae that were not crowded did not trigger a significant response (Figure 3b). These results also support our original prediction, that ovipositing females respond to semiochemocals released by crowded *C. longiareolata* larvae, indicating a habitat with high competition [34].

The distribution of *C. pipiens* egg rafts was not significantly affected by the *C. longiareolata* cues (Figure 2c) or larvae (Figure 3c). The lack of response shown by this species to the presence of *C. longiareolata* larvae may be associated with the cosmopolitan distribution of this species. The global distribution of *C. laticinctus* is contained within that of *C. longiareolata* [20]. Furthermore, the larvae of these two species often co-occur in recently filled freshwater pools who are considered as the preferred oviposition site for both species [20,24,42]. *Culex pipiens* on the other hand, are characterized by global distribution and by their ability to inhabit a very wide variety of water sources [20]. As such, this species may be less likely to identify a specific larval competitor such as *C. longiareolata* that only occupies some of its preferred breeding sites.

In conclusion, the two field experiments show a distinct, species-specific reaction of ovipositing gravid mosquito females to mosquito larvae and their released chemical signals. This observed reaction is in response to semiochemicals associated with larval *C. longiareolata*, indicating increasing larval density. The species- and density-specific responses of the three mosquitoes strongly support the idea that ovipositing mosquito females can detect larval competitors via chemical signals. Future studies should also look into the effects of semiochemicals released by interspecific eggs and egg rafts, and their chemical signatures.

## Figures and Tables

**Figure 1 insects-15-00364-f001:**
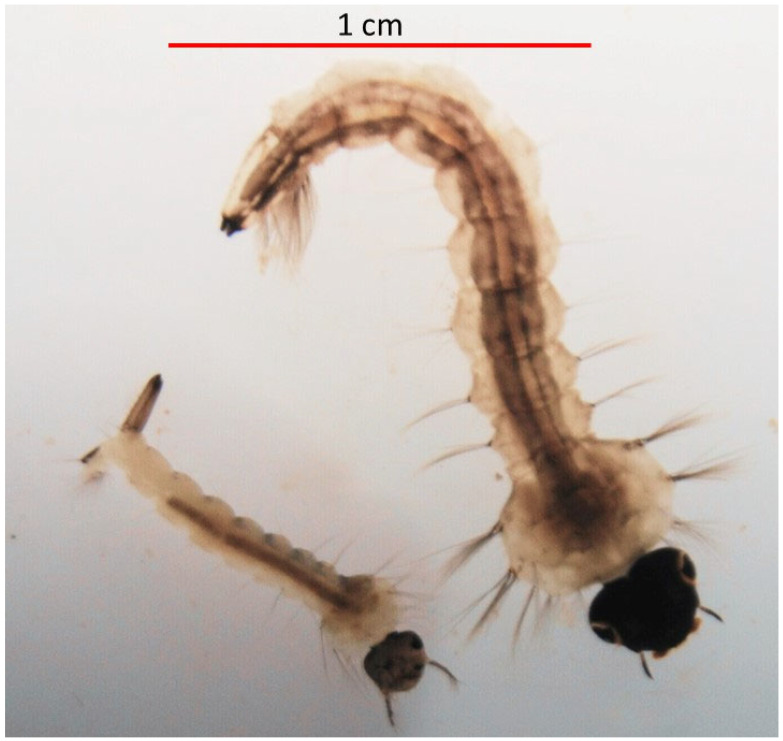
Fourth instar larval: *Culiseta longiareolata* (top) and *Culex laticinctus*. Captured using Nikon, Singapore, (SMZ18) fluorescence-dissecting microscope connected to a Nikon DS-Fi3 camera.

**Figure 2 insects-15-00364-f002:**
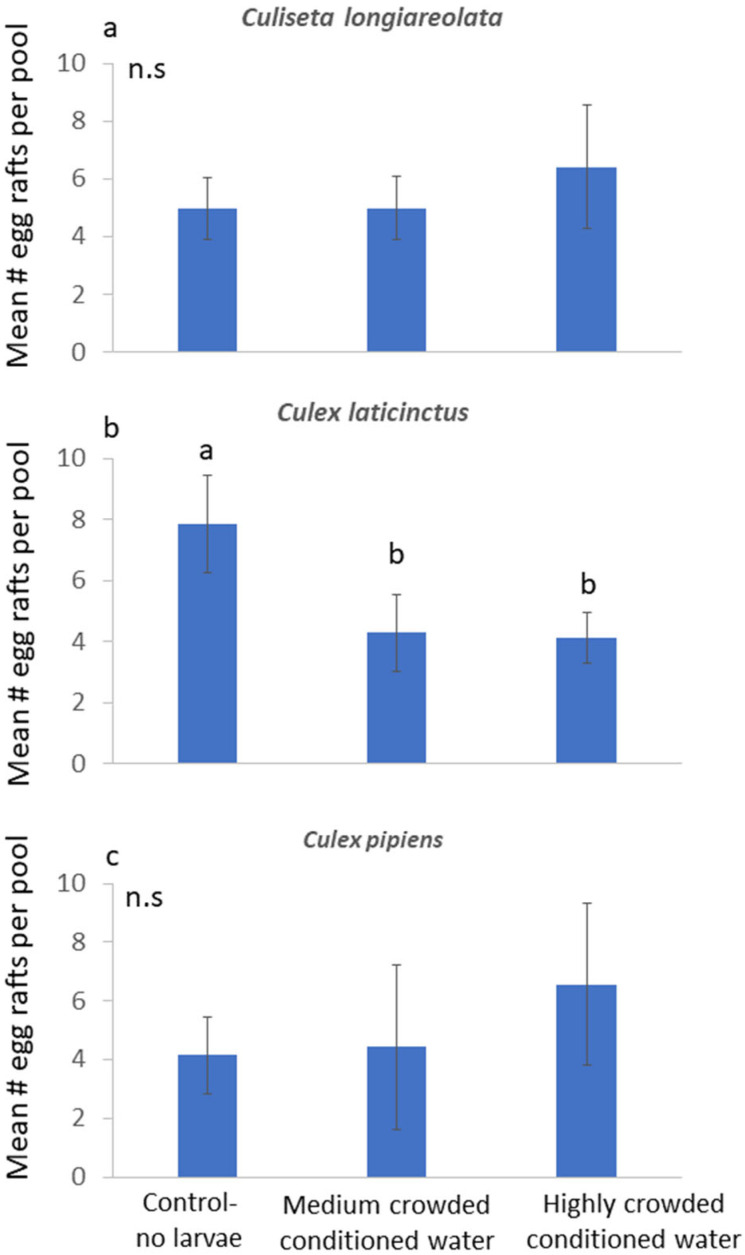
Effect of *Culiseta logiareolata* larvae conditioned water on ovipositing: (**a**) *Culiseta logiareolata*; (**b**) *Culex laticinctus*; and (**c**) *Culex pipiens* oviposition. n = 7 per treatment. Error bars stand for ±1 SE. Different letters indicate treatments that are significantly different based on post hoc comparisons. n.s.: not significant.

**Figure 3 insects-15-00364-f003:**
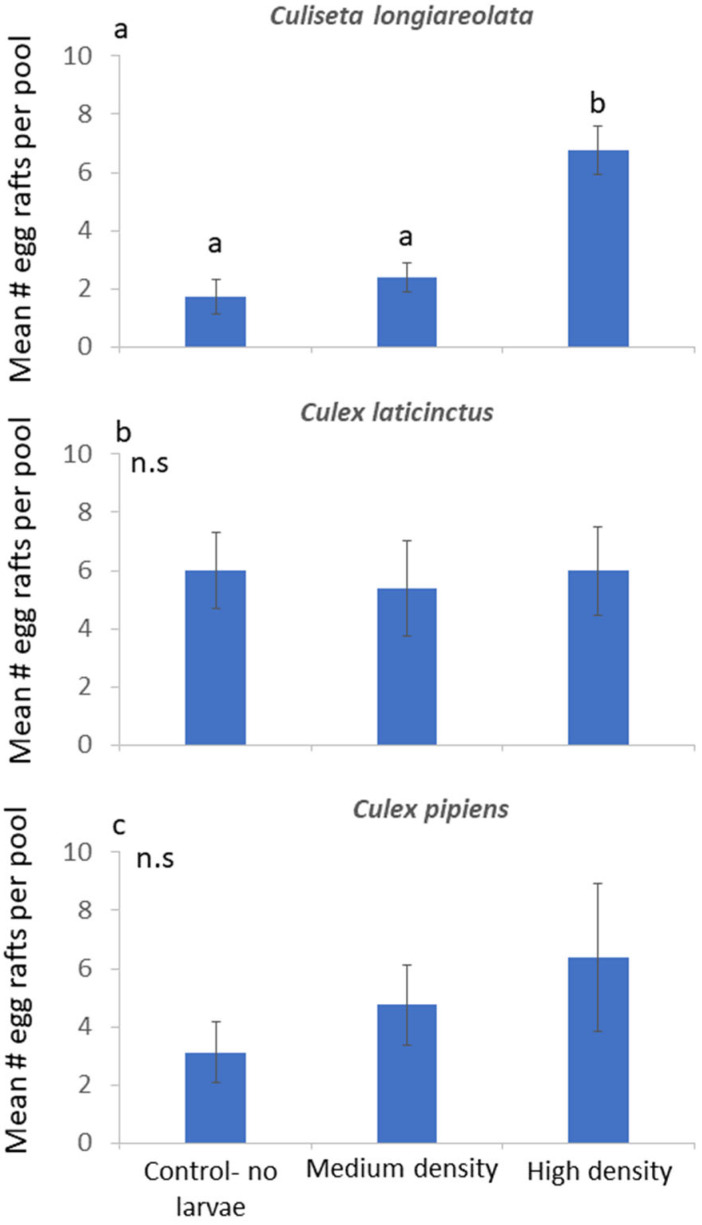
Effect of *Culiseta logiareolata* larvae on ovipositing: (**a**) *Culiseta logiareolata*; (**b**) *Culex laticinctus*; and (**c**) *Culex pipiens* oviposition. n = 8 per treatment. Error bars stand for ±1 SE. Different letters indicate treatments that are significantly different based on post hoc comparisons. n.s.: not significant.

## Data Availability

Data are available upon request from the authors.

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
