# Peer review of "Mosquito Egg Raft Distribution Is Affected by Semiochemicals: Indication of Interspecific Competition"

_insects, 2024, doi:10.3390/insects15050364_

Round 1

Reviewer 1 Report

Comments and Suggestions for Authors

A very interesting well written manuscript.  Only a few minor questions/comments.  Line 117, what was the source of the finely grounded fish flakes?  Do the authors think the results would have been different if they used water into which egg rafts of different densities were used instead of larvae densities.  Don't some species of mosquitoes deposit droplets containing semiochemicals during oviposition?  

Why were the pools emptied and refilled every five days?  Do the authors think that some semiochemicals could have been impregnated into the plastic of the pools and released during refilling? Any idea what the composition of semiochemicals produced by the larvae was?  Any ideas on why there was such low oviposition in the one block which was excluded from the statistical analysis?

Line 213, district should be distinct.

Author Response

Comment

Response

Line 117, what was the source of the finely grounded fish flakes?

Data added- line 120

Do the authors think the results would have been different if they used water into which egg rafts of different densities were used instead of larvae densities.

Indeed an interesting subject for future studies. We mentioned this in lines 224-226.

Why were the pools emptied and refilled every five days?

We wanted to look at a system with as little environmental noise as possible. We address this issue in lines 123-124

Do the authors think that some semiochemicals could have been impregnated into the plastic of the pools and released during refilling?

Always a possibility, but highly unlikely in such a short time period.

Any idea what the composition of semiochemicals produced by the larvae was?

Another interesting subject for future studies. We mentioned this in lines 224-226.

Any ideas on why there was such low oviposition in the one block which was excluded from the statistical analysis?

No, but this is not uncommon in field studies. The block design allows us to deal with it by removing from analysis, blocks with unusual values.

Line 213, district should be distinct.

Changed

Reviewer 2 Report

Comments and Suggestions for Authors

Have you done any collection and chemical analysis for pheromone of Culex eggs rafts? There are several reports about this. 

The number of egg rafts that have been used for comparison of the interspecies reaction seems reasonable. If you can detect the chemicals, it will be great to answer many questions. 

Probably it is better to change the title to using the number of egg rafts for justification.

Provide more detailed information for Fig 1 caption.

Provide the GPS location for your experimental site/location.

You did not mention anything about Culex pipiens, but you have all data about Culex pipiens in your figures.

Comments on the Quality of English Language

It is good.

Author Response

Comment

Response

Have you done any collection and chemical analysis for pheromone of Culex eggs rafts? There are several reports about this.

A very interesting subject for further research. We added it in lines 224-226

Probably it is better to change the title to using the number of egg rafts for justification

We changed the title

Provide more detailed information for Fig 1 caption.

Added

Provide the GPS location for your experimental site/location

Added on line 107

You did not mention anything about Culex pipiens, but you have all data about Culex pipiens in your figures

We argue (lines 213-219) that the lack of response by this species may be explained by its wider niche breadth.

Round 2

Reviewer 2 Report

Comments and Suggestions for Authors

Thanks for answering my questions.

Comments on the Quality of English Language

minor editing

Author Response

This reviewer did not have additional comments.